# Solar-blind ultraviolet-C persistent luminescence phosphors

Xianli Wang[1,4], Yafei Chen[2,4], Feng Liu[3] & Zhengwei Pan[2✉]

Visible-light and infrared-light persistent phosphors are extensively studied and are being used as self-sustained glowing tags in darkness. In contrast, persistent phosphors for higher-energy, solar-blind ultraviolet-C wavelengths (200–280 nm) are lacking. Also, persistent tags working in bright environments are not available. Here we report five types of $Pr^{3+}$-doped silicates (melilite, cyclosilicate, silicate garnet, oxyorthosilicate, and orthosilicate) ultraviolet-C persistent phosphors that can act as self-sustained glowing tags in bright environments. These ultraviolet-C persistent phosphors can be effectively charged by a standard 254 nm lamp and emit intense, long-lasting afterglow at 265–270 nm, which can be clearly monitored and imaged by a corona camera in daylight and room light. Besides thermal-stimulation, in bright environments, photo-stimulation also contributes to the afterglow emission and its contribution can be dominant when ambient light is strong. This study expands persistent luminescence research to the ultraviolet-C wavelengths and brings persistent luminescence applications to light.

---

[1] College of Engineering, University of Georgia, Athens, GA 30602, USA. [2] Center for Integrative Petroleum Research, College of Petroleum Engineering and Geosciences, King Fahd University of Petroleum and Minerals, Dhahran 31261, Kingdom of Saudi Arabia. [3] Key Laboratory for UV-Emitting Materials and Technology of Ministry of Education, Northeast Normal University, 130024 Changchun, China. [4]These authors contributed equally: Xianli Wang, Yafei Chen. ✉email: zhengwei.pan@gmail.com

Ultraviolet-C (UVC) spectral band refers to the spectrum of light between 200 and 280 nm. Solar radiation in this spectral interval does not reach the Earth surface due to ozone absorption. Therefore, this spectral domain is also called "solar blind". Artificial UVC radiation sources play an important role within several markets worldwide today. One primary market is disinfection and photochemical purification of water, owing to the high spectral overlap of the UVC range and the germicidal effectiveness curve (~220–280 nm, peaked at 265 nm)[1–4]. Other applications of UVC radiation include photodynamic therapy[5], photocatalysis[6], photo-curing of polymers and resins[7], etc. Moreover, the absence of solar UVC radiation in the lower atmosphere produces zero background conditions[8,9], enabling artificial UVC radiation, such as electric discharges in power utilities, to be monitored and recorded using a UVC corona camera with high-contrast and identical effect in daytime and at night[10,11].

The market for UVC radiation sources is currently dominated by low-pressure mercury vapor discharge lamps that emit at 254 nm. For disinfection application, however, the UVC sources had better emitting at around 265 nm—the most effective germicidal wavelength. Accordingly, efforts have been spent to develop UVC phosphors that can downshift higher-energy UVC radiation (e.g., low-pressure mercury lamp emitting at 254 nm) or vacuum UV radiation (e.g., xenon excimer lamp emitting at 172 nm)[12] to the more effective germicidal wavelengths. In developing UVC phosphors, $Pr^{3+}$ is the favorite activator[13]. The UVC emission of the $Pr^{3+}$-activated UVC phosphors is dominated by broad, parity allowed $Pr^{3+}$ $4f^{1}5d^{1} \rightarrow 4f^{2}$ interconfigurational transitions (Fig. 1a). To ensure the occurrence of $Pr^{3+}$ $4f^{1}5d^{1} \rightarrow 4f^{2}$ transitions in the UVC in a solid, two general conditions are required: a small Stokes shift of less than about 3000 cm$^{-1}$ (0.37 eV) and an appropriate energy location of the first (lowest energy) $Pr^{3+}$ $4f^{2} \rightarrow 4f^{1}5d^{1}$ excitation transition[14–18], which are associated with the compositions and crystal structures of the host lattices. Under these conditions, the nonradiative relaxation of the $Pr^{3+}$ $4f^{1}5d^{1}$ level to the lower $4f^{2}$ ($^{3}P_{J}$, $^{1}I_{6}$, $^{1}D_{2}$) levels is minimized; otherwise, crossing of the $4f^{1}5d^{1}$ level with

the lower $4f^{2}$ levels will occur and, as a result, sharp line $4f^{2} \rightarrow 4f^{2}$ intraconfigurational emission transitions for visible and/or infrared-light emission will dominate (Fig. 1a). Up to now, tens of $Pr^{3+}$-activated UVC phosphors were developed with the commonly used hosts including fluorides (e.g., $LiYF_{4}$), phosphates (e.g., $YPO_{4}$), borates (e.g., $YBO_{3}$), and aluminates (e.g., $YAlO_{3}$).

The $Pr^{3+}$-activated UVC phosphors developed by far are almost all photoluminescence. For some applications, such as surveillance, it is highly desirable that the UVC materials can emit long-lasting, self-sustained UVC luminescence without the need of constant external excitation, a special optical phenomenon called persistent luminescence or afterglow[19]. In the past two decades, the world witnessed rapid progress in persistent phosphors research and applications, and hundreds of persistent phosphors with emission wavelengths in the visible[20] and infrared[21] spectral regions were developed. The visible-infrared persistent phosphors are being widely used as self-sustained night-vision tags or imaging probes in dark environments; their tagging applications in bright environments such as in daylight and room light, however, are encumbered because of the high background luminescence from the Sun or artificial lighting sources[8,9]. In stark contrast to the evolving progress in the visible-infrared spectral regions, the research at the other end of the spectrum—the shorter-wavelength UVC region remains very quiet; only one $Pr^{3+}$-doped fluoride-based UVC persistent phosphor ($Cs_{2}NaYF_{6}$: $Pr^{3+}$) with a afterglow duration of ~2 h after X-ray radiation was recently reported[22]. For broad applications of UVC persistent phosphors, however, the materials had better be excitable and chargeable by widely available and less harmful light sources, e.g., a standard 254 nm mercury discharge lamp, and exhibiting a much longer persistence time, e.g., >10 h.

Achieving UVC persistent luminescence in a $Pr^{3+}$-activated phosphor by a "low-energy" 254 nm lamp excitation is very challenging, which puts forward rigorous requirements to the host lattice (e.g., host composition and crystal structure). The phosphor host needs to be capable of creating appropriate energy traps, which should locate at significantly high-energy position (just below $Pr^{3+}$ ion's lowest energy $4f^{1}5d^{1}$ level) and yet should

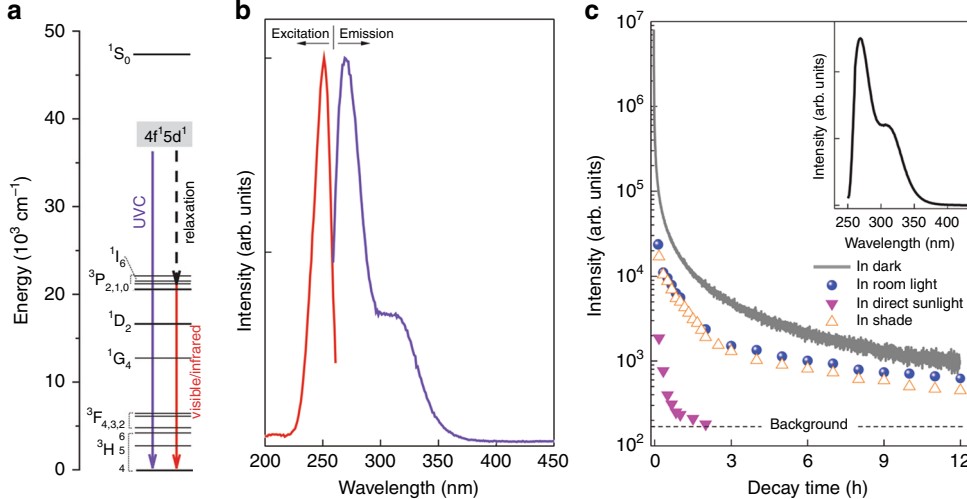

**Fig. 1 Photoluminescence and persistent luminescence of Ca₂Al₂SiO₇:Pr³⁺ phosphor. a** Energy level scheme of $Pr^{3+}$ $4f^{2} \rightarrow 4f^{1}5d^{1}$ configuration and the possible optical transitions. **b** Normalized photoluminescence excitation and emission spectra at room temperature. The emission spectrum is acquired under 251 nm light excitation and the excitation spectrum is obtained by monitoring 268 nm emission. **c** Ultraviolet-C persistent luminescence decay curves monitored at 268 nm in darkness (i.e., in spectrofluorometer sample chamber), direct sunlight, indirect sunlight (i.e., the shade of a building), and room light. The samples were pre-irradiated by a 254 nm lamp for 2 min. For in darkness, the decay curve was recorded continuously for 12 h. For in direct sunlight, indirect sunlight, and room light, the samples were taken back to the spectrofluorometer to measure the remaining persistent luminescence intensity at every 10 min in the first 1 h and at variable time interval (10 min, 30 min, or 1 h) in the rest of decay. Inset shows the persistent luminescence emission spectrum acquired at 1 h decay in darkness. Source data for **b**, **c** are provided as a Source Data file.

be effectively filled by 254 nm light. The host should also be capable of creating a suitably strong crystal field environment to ensure a small Stokes shift that is required for efficient $Pr^{3+}$ $4f^15d^1 \rightarrow 4f^2$ transitions[14–18]. During the course of searching $Pr^{3+}$-activated UVC persistent phosphors, we firstly tested some prominent $Pr^{3+}$-activated UVC photoluminescence phosphors, including $YPO_4:Pr^{3+}$, $YBO_3:Pr^{3+}$, and $Lu_3Al_5O_{12}:Pr^{3+}$, but did not observe $Pr^{3+}$ UVC afterglow in them, suggesting the lack of appropriate energy traps in these phosphors. However, we found that highly coordinated silicate-containing compounds, particularly the silicates used for $Ce^{3+}$-activated persistent phosphors or mechanoluminescence phosphors (e.g., $Ca_2Al_2SiO_7:Ce^{3+}$ (refs.[23–25]) and $Lu_2SiO_5:Ce^{3+}$ (refs.[26,27])), are the suitable hosts for $Pr^{3+}$ UVC afterglow after 254 nm light irradiation. We then identified five types, more than ten kinds of silicates for $Pr^{3+}$ UVC afterglow, including melilites, $M_2Al_2SiO_7$ ($M =$ Ca, Sr); cyclosilicates, $M_3Y_2Si_6O_{18}$ ($M =$ Ca, Sr, or Sr + Ca); silicate garnets, $M_3X_2Si_3O_{12}$ ($M =$ Ca, Sr; $X =$ Al, Al + Ga, or Al + Y); oxyorthosilicates, $R_2Si_xO_{3+2x}$ ($R =$ Lu, Y; $x = 1$ or 2); and orthosilicates, $LiRSiO_4$ ($R =$ Y, Lu).

Our study shows that these $Pr^{3+}$-doped silicate UVC persistent phosphors can be effectively charged by a standard 254 nm lamp and emit intense, long-lasting (~10 h) afterglow at 265–270 nm. Benefited from the zero solar UVC background on the Earth surface, the UVC afterglow can be clearly monitored and imaged by a corona camera in daylight and room light, enabling the UVC persistent phosphors to act as self-sustained, solar-blind glowing tags in bright light. Besides thermal stimulation, in bright environments, photostimulation caused by the ambient light also contributes to the afterglow emission and its contribution can be dominant when the ambient light is strong. In this article, we focus our discussion on the melilite-structured $Ca_2Al_2SiO_7:Pr^{3+}$ persistent phosphor. The results of other types of silicates are given in Supplementary Information.

## Results

**Structure and photoluminescence of $Ca_2Al_2SiO_7:Pr^{3+}$ phosphor**. The as-synthesized $Ca_2Al_2SiO_7:Pr^{3+}$ compound has the melilite structure (Supplementary Fig. 1). In melilite $Ca_2Al_2SiO_7$, the $Ca^{2+}$ ions are sandwiched between the layers of $AlO_4$ and $SiO_4$ tetrahedrons alternating along the $c$ axis and are eightfold coordinated[27,28]. Each $Ca^{2+}$ ion is bonded to four nearest neighbor $O^{2-}$ ligand ions in both $AlO_4$ layer and $SiO_4$ layer, and the thus formed four $Ca^{2+}$ complexes in a unit cell are structurally equivalent[25]. In $Ca_2Al_2SiO_7:Pr^{3+}$ crystal, trivalent $Pr^{3+}$ (1.126 Å) ions substitute for smaller, divalent $Ca^{2+}$ (1.12 Å) ions; therefore, the doped $Pr^{3+}$ ions are eightfold coordinated. Such highly coordinated, smaller, and charge-imbalanced cation sites can create a suitably strong crystal field for $Pr^{3+}$ ions[29], by which a small Stokes shift and therefore an efficient $Pr^{3+}$ $4f^15d^1 \rightarrow 4f^2$ interconfigurational transition for UVC emission is highly likely to occur. Moreover, the cation size mismatch and charge imbalance are expected to create more effective energy traps (e.g., oxygen vacancies) around $Pr^{3+}$ ions[24,25,30], which are necessary for a good persistent phosphor. However, it is worth noting here that the current characterization techniques are insufficient for us to understand and determine the exact nature of the trap states and how the traps are formed in $Ca_2Al_2SiO_7:Pr^{3+}$ phosphor.

Figure 1b shows the normalized photoluminescence excitation and emission spectra of $Ca_2Al_2SiO_7:Pr^{3+}$ phosphor at room temperature. The excitation spectrum monitored at 268 nm emission consists of one main excitation band (~220–260 nm) peaking at 251 nm (39,841 $cm^{-1}$) and one weak band peaking at about 206 nm (48,544 $cm^{-1}$). The main excitation band can be assigned to the lowest energy $Pr^{3+}$ $4f^2 \rightarrow 4f^15d^1$ excitation

transition. The excitation spectrum suggests that the $Ca_2Al_2SiO_7:Pr^{3+}$ phosphor can be effectively excited by a standard 254 nm mercury discharge lamp. Under excitation at 251 nm, the phosphor exhibits a strong broadband emission peaking at 268 nm (37,313 $cm^{-1}$) and a weak shoulder band peaking at about 314 nm (31,847 $cm^{-1}$), which can be attributed to the $Pr^{3+}$ $4f^15d^1 \rightarrow 4f^2$ emission transitions. No apparent emission in the visible region is observed in $Ca_2Al_2SiO_7:Pr^{3+}$. Moreover, Fig. 2b clearly shows that the Stokes shift of $Pr^{3+}$ emission in $Ca_2Al_2SiO_7:Pr^{3+}$ is very small, only about 2528 $cm^{-1}$, which satisfies the requirement (<3000 $cm^{-1}$) for achieving $Pr^{3+}$ UVC emission in solids[14–18].

**Ultraviolet-C persistent luminescence**. When the excitation was ceased, UVC persistent luminescence emission at 268 nm was obtained. The gray curve in Fig. 1c shows the persistent luminescence decay curve of a $Ca_2Al_2SiO_7:Pr^{3+}$ disc monitored at 268 nm at room temperature in darkness after irradiation by a standard, 4-W 254 nm lamp for 2 min. The data were recorded as a function of persistent luminescence intensity versus time and the recording lasted for 12 h. The persistent luminescence intensity decreases quickly in the first 1 h and then decays slowly. After 12 h decay, the persistent luminescence intensity is still over one order of magnitude higher than the background signal. The UVC persistent luminescence power intensities at 1 s and 10 s decay instants are estimated to be about 10.9 mW m$^{-2}$ and 4.8 mW m$^{-2}$, respectively (see "Methods", Supplementary Note 1 and Supplementary Fig. 2). The profile of the persistent luminescence emission spectrum (inset in Fig. 1c) is identical to that of the photoluminescence emission spectrum (Fig. 1b), indicating that the UVC persistent luminescence originates from the $Pr^{3+}$ $4f^15d^1 \rightarrow 4f^2$ emission transitions. The observation of $Pr^{3+}$ UVC afterglow in $Ca_2Al_2SiO_7:Pr^{3+}$ indicates that the energy traps in the material can be effectively filled by 254 nm light excitation and that the energy traps are located at appropriate energy positions so that they can efficiently capture the electrons from the $Pr^{3+}$ $4f^15d^1$ state during the excitation and release the electrons back to the $4f^15d^1$ state due to ambient thermal stimulation after the excitation is ceased. Such electron trapping and detrapping processes in the $Ca_2Al_2SiO_7:Pr^{3+}$ persistent phosphor are expected to share similar mechanism to other persistent phosphors such as the $Cr^{3+}$-activated near-infrared persistent phosphors[21]. The occurrence of long $Pr^{3+}$ UVC afterglow in $Ca_2Al_2SiO_7:Pr^{3+}$ also indicates that there exist highly dense, continuously distributed energy traps in the material. Indeed, thermoluminescence measurements carried out on $Ca_2Al_2SiO_7:Pr^{3+}$ samples underwent different decay time (from 1 min to 24 h) at room temperature in darkness reveal that the thermoluminescence band is very broad, covering from 25 °C to ~280 °C (Supplementary Fig. 3). The thermoluminescence experiments also show the trap emptying process during the decay; that is, the shallow traps are quickly emptied in the first few hours, followed by the slowly emptying of deep traps, resulting in the thermoluminescence band maximum shifting from ~115 °C at 1 min decay to ~221 °C at 24 h decay. Remarkably, a considerable amount of trapped electrons were still not liberated even after 24 h decay at room temperature in darkness.

**Ultraviolet-C imaging in daylight and room light**. The realization of long-lasting UVC afterglow in $Ca_2Al_2SiO_7:Pr^{3+}$ enables us to explore some new applications that cannot be achieved by the visible and infrared persistent phosphors. One remarkable application is tagging in bright environments. By taking advantage of the zero solar UVC background on the Earth surface, we use a UVC corona camera to demonstrate the unique capability

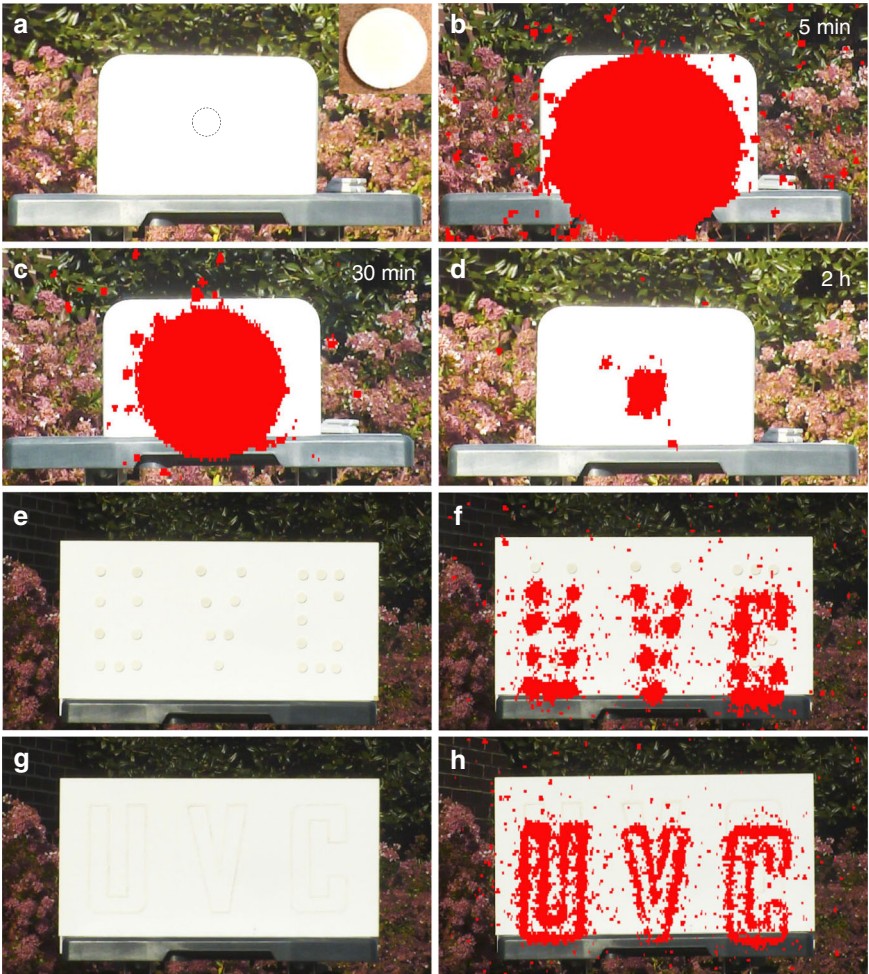

**Fig. 2 Ultraviolet-C images of decaying Ca$_2$Al$_2$SiO$_7$:Pr$^{3+}$ discs in direct sunlight. a** Visible image of a 50 mm diameter white-color disc adhered on a white foam board. The position of the disc is indicated by a dash-line circle. Inset shows the white disc on a darker background. **b–d** Ultraviolet-C (UVC) images of the disc taken at 5 min, 30 min, and 2 h decay using a corona camera. **e** Visible image of letters "U", "V", and "C" made of 20 mm diameter discs on a white wood board. **f** UVC image of the disc letters taken at 2 min decay. **g** Visible image of hollow letters "U", "V", and "C" written using UVC paint on a white wood board. **h** UVC image of the paint letters taken at 2 min decay. In these imaging experiments, the samples were pre-irradiated by a 254 nm lamp for 2 min. The camera was located about 8 m away from the samples. The UVC radiation is represented by red color.

of the Ca$_2$Al$_2$SiO$_7$:Pr$^{3+}$ UVC persistent phosphor as self-luminescing, solar-blind tags for high-contrast, and sensitive surveillance in bright indoor and outdoor environments. The tested samples include Ca$_2$Al$_2$SiO$_7$:Pr$^{3+}$ ceramic discs with diameters of 50 and 20 mm and UVC paints made of microscale Ca$_2$Al$_2$SiO$_7$:Pr$^{3+}$ powder in acrylic polyurethane varnish. The camera is an Ofil DayCor Luminar HD corona camera, which can take bi-spectral UVC (for 240–280 nm)-visible images. Figure 2a is a visible image taken in direct sunlight showing an uncharged 50 mm diameter white-color Ca$_2$Al$_2$SiO$_7$:Pr$^{3+}$ disc adhered onto a white foam board before excitation (inset shows the disc). Under direct sunlight, the white disc is basically indistinguishable from the white board for unaided eye from about 8 m away. After irradiation by a 254 nm lamp for 2 min, however, intense UVC afterglow radiation from the Ca$_2$Al$_2$SiO$_7$:Pr$^{3+}$ disc can be clearly detected and imaged by the corona camera for up to 2 h in direct sunlight (Fig. 2b–d). (It is worth noting here that, as that will be discussed later, the UVC radiation signal of a decaying Ca$_2$Al$_2$SiO$_7$:Pr$^{3+}$ disc in direct sunlight is contributed by two types of luminescence: the normal persistent luminescence due to ambient heat stimulation and the photostimulated luminescence (PSL) due to sunlight stimulation, and the latter is dominant). The size of the UVC radiation corona in the first 1 h of decay (Fig. 2b, c) is

much larger than the real size of the disc, due to the intense initial luminescence and the scattering of the UVC light with the molecules and aerosols present in the atmosphere[31]. After more than 2 h of decay in direct sunlight, the UVC radiation becomes extinguished to the corona camera. Similar imaging results in direct sunlight were also observed on 20 mm diameter discs (Fig. 2e, f) and paint (Fig. 2g, h).

To further demonstrate the surveillance capability of Ca$_2$Al$_2$SiO$_7$:Pr$^{3+}$ persistent phosphor in direct sunlight, we conducted UVC imaging experiments by placing the charged 50 mm diameter Ca$_2$Al$_2$SiO$_7$:Pr$^{3+}$ discs in different outdoor environments, including on tree trunk (Supplementary Fig. 4a–d), in bush (Supplementary Fig. 4e, f), on brick wall (Supplementary Fig. 4g, h), in water (Supplementary Fig. 5), and at long distance (up to ~50 m) from the camera (Supplementary Fig. 6). Sharp UVC images were consistently obtained in direct sunlight in these tests.

We also visually evaluated the decay performance of Ca$_2$Al$_2$SiO$_7$:Pr$^{3+}$ persistent phosphor in room light. Figure 3a–d show the UVC images of a 50 mm diameter Ca$_2$Al$_2$SiO$_7$:Pr$^{3+}$ disc at 5 min to 24 h decay in room light (UVC images for 20 mm diameter discs and paint in room light are given in Supplementary Fig. 7a, b and Supplementary Fig. 7d, e, respectively). After 24 h decay in room light, the UVC radiation can still be clearly imaged

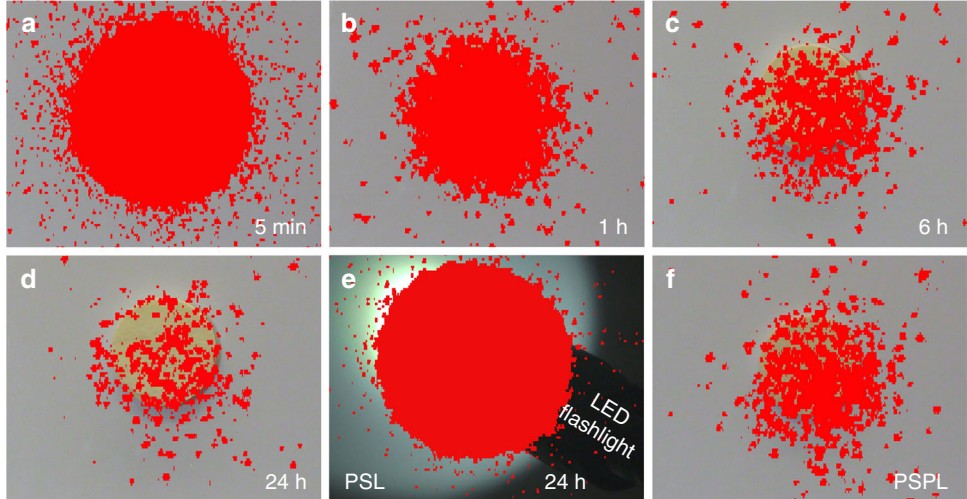

**Fig. 3 Ultraviolet-C images of a decaying Ca$_2$Al$_2$SiO$_7$:Pr$^{3+}$ disc in room sunlight. a–d** Ultraviolet-C (UVC) images of a 50 mm diameter disc taken at different decay time (5 min to 24 h) using a corona camera. The disc was pre-irradiated by a 254 nm lamp for 2 min. **e** UVC photostimulated luminescence (PSL) image of the 24 h decayed disc being irradiated by a 3800 Lumen white LED flashlight. The irradiation lasted for 10 s. **f** UVC photostimulated persistent luminescence (PSPL) image taken at 1 min after ceasing the flashlight irradiation. The UVC radiation is represented by red color. The camera was located about 5 m away from the disc.

by the corona camera (Fig. 3d). Finally, the UVC images taken in darkness (Supplementary Fig. 7c, f) have the same clarity as those taken in room light (Supplementary Fig. 7b, e) and in direct sunlight (Fig. 2f, h).

**Photostimulated luminescence in the ultraviolet-C.** In imaging experiments, we noted that the decay performance of Ca$_2$Al$_2$SiO$_7$:Pr$^{3+}$ persistent phosphor was significantly influenced by the ambient lighting conditions. For instance, the imaging duration in direct sunlight is an order of magnitude shorter than that in room light (~2 h vs. ~24 h); the radiation intensity at the same 5 min decay instant in direct sunlight is much more intense than that in room light (Fig. 2b vs. Fig. 3a). The influences of ambient light on the decay were also quantitatively evaluated using spectral method by measuring the remaining afterglow intensity as a function of exposure time in direct sunlight, indirect sunlight (i.e., in shade of a building), and room light (fluorescence lamp light), and comparing them with the decay in darkness, as shown by the four decay curves in Fig. 1c. (Note that the remaining afterglow intensities were measured in darkness in the spectrofluorometer chamber, which are lower than those being in light but can quantitatively reflect the level of energy remained in the samples). In darkness, the sample decays slowly and exhibits the longest afterglow duration; in direct sunlight, by contrast, the opposites are true. Remarkably, in outdoor but in shade, i.e., in indirect sunlight, the decay can last for much longer than that in direct sunlight, but is just slightly shy to the case in room light. The faster decay in brighter ambient lighting conditions indicates that the ambient light can boost the release of the trapped energy in the phosphor.

To illustrate the effects of ambient light on the release of the trapped energy, we illuminated a 12 h decayed (in darkness) Ca$_2$Al$_2$SiO$_7$:Pr$^{3+}$ disc using monochromatic light between 300 and 700 nm in 50 nm steps for 30 s, and recorded the 268 nm emission intensity during and after the illumination, as shown in Fig. 4a. After 12 h of natural decay in darkness, the persistent luminescence emission intensity of the disc becomes very weak. However, upon the 300–700 light illumination, the UVC emission intensity of the decayed sample increases sharply, particularly for the 300–550 nm light illumination, and exhibits a direct relationship with the photon energy. Such an intense UVC emission is caused by the stimulated release of the remaining energy in the

decaying sample by the illumination light, a well-known optical phenomenon called as PSL which is the fundamental property of storage phosphors[32,33]. The PSL phenomenon indicates that, when ambient light (sunlight or room light) exists, the UVC emission of a decaying Ca$_2$Al$_2$SiO$_7$:Pr$^{3+}$ sample is no longer a sole thermally stimulated persistent luminescence as the case in darkness; instead, the UVC signal is contributed by two types of luminescence signal: the normal persistent luminescence signal due to ambient heat stimulation and the PSL signal due to ambient light stimulation. The contribution from the PSL increases as the light energy or light intensity increases, and it can be dominant if the ambient light is strong enough. Specifically, in direct sunlight, the solar spectrum contains intense 300–700 nm UV–visible light (the outdoor light level is ~100,000 lx on a clear sky midday), which, as shown in Fig. 4a, can strongly boost the release of the stored energy in the phosphor and produce strong PSL signal; therefore, the decay of Ca$_2$Al$_2$SiO$_7$:Pr$^{3+}$ phosphor in direct sunlight can only last for ~2 h. In shade, the indirect sunlight also contains UV and visible components, but the light level is only about one-fifth of the direct sunlight, resulting in a moderate stimulation effect. The room light does not contain UV component and its intensity is less than 1% of the direct sunlight; therefore, the room light produces the least stimulation effect among the three lighting conditions. Overall, the PSL effect discussed above well explains the ambient light dependent decay behaviors observed in the imaging and spectral experiments. The PSL effect also suggests that when a persistent phosphor is used in bright light, the stimulation effect of the light must be taken into consideration.

**Photostimulated persistent luminescence in the ultraviolet-C.** Besides producing strong PSL signal during illumination, the short illumination by the 300–700 nm "low-energy" light (relative to the high-energy 254 nm excitation light) can also significantly increase the persistent luminescence intensity of the decaying sample (lower inset of Fig. 4a) and the effectiveness increases with the energy of the irradiation light (upper inset of Fig. 4a). This enhanced persistent luminescence due to low-energy light illumination is referred to as photostimulated persistent luminescence (PSPL), which is a common optical phenomenon in persistent phosphors[34–36]. It is worth noting that for the UVC

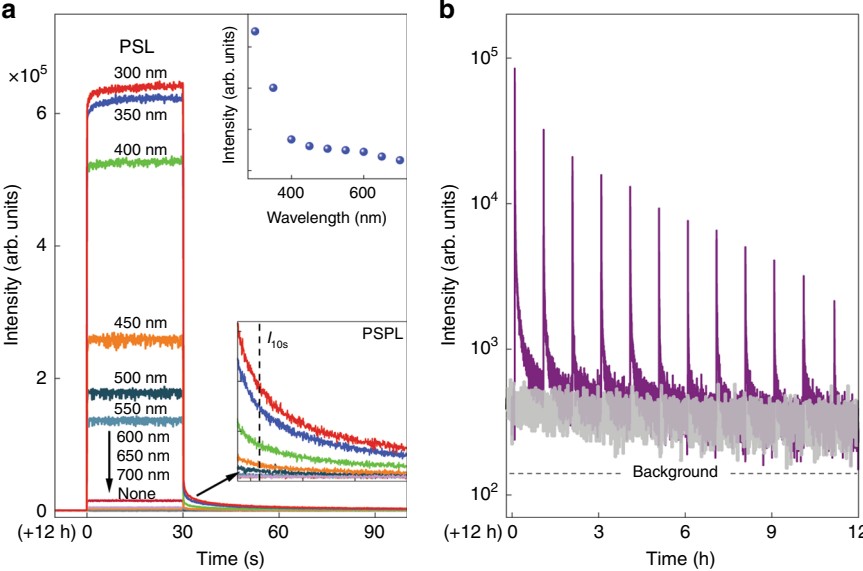

**Fig. 4 Photostimulation effects on decayed $Ca_2Al_2SiO_7$:$Pr^{3+}$ discs. a** Photostimulated luminescence (PSL) spectra and photostimulated persistent luminescence (PSPL) decay curves of a 12 h decayed disc (in darkness; pre-irradiated by a 254 nm lamp for 2 min) under and after 300–700 nm monochromatic light stimulation for 30 s. The monitoring wavelength is 268 nm. The lower inset shows the enlarged PSPL decay curves. The PSPL intensity at time of 10 s ($I_{10s}$) after the stoppage of the stimulation was used to plot the PSPL intensity as a function of stimulation wavelengths, as shown in the upper inset. **b** Repeated PSPL decay curves obtained on a 12 h decayed disc (in darkness; pre-irradiated by a 254 nm lamp for 2 min). The monitoring wavelength is 268 nm. The decayed disc was irradiated by a 3800 Lumen white LED flashlight for 10 s at every 1 h for a total of 12 times to elicit the ultraviolet-C PSPL signals. For comparison, the gray curve shows the continuous decay for another 12 h in darkness (i.e., without the LED flashlight irradiation). Source data are provided as a Source Data file.

PSPL in $Ca_2Al_2SiO_7$:$Pr^{3+}$, the essential excitation source is the 254 nm lamp. The subsequent low-energy 300–700 nm light illumination cannot fill the energy traps, but it can liberate some electrons in the deep traps to the $Pr^{3+}$ $4f^15d^1$ level. While most of the released electrons radiatively return to the ground state via $Pr^{3+}$ $4f^15d^1 \rightarrow 4f^2$ transitions producing strong UVC PSL signal during the illumination, some electrons refill the depleted shallow traps, resulting in enhanced UVC PSPL emission after the illumination is turned off[34]. The redistribution of electrons from the deep traps to the shallow traps due to low-energy light illumination is revealed by the black dots-line thermoluminescence curve in Supplementary Fig. 3, where a 12 h decayed $Ca_2Al_2SiO_7$:$Pr^{3+}$ disc was illuminated by a 3800 Lumen white LED flashlight for 20 s. Since a short-time, low-energy light illumination can photo-liberate only partial electrons in the deep traps, the PSPL process can be carried out for multiple times (>10 times) over a long period (up to tens of hours) on a decaying disc until all energy traps in the disc are emptied (Fig. 4b). This repeated PSPL capability has important implication for some applications in darkness, particularly for in vivo medical applications, because once the pre-charged material (as nanoparticles) is inside a living object, it is not possible to recharge it using 254 nm light, but the stored energy in the material can be maximally released by high-tissue-penetrating visible light from, e.g., a white LED flashlight[37].

The PSL and PSPL phenomena were also verified by the imaging method, as shown in Fig. 3e, f, respectively, where a 24 h decayed disc in room light was illuminated by a 3800 Lumen white LED flashlight for 10 s (see also Supplementary Fig. 8 for PSL and PSPL on a 48 h decayed disc in room light). The thus generated PSL signal (Fig. 3e) is as intense as that at 5 min decay (Fig. 3a) and the PSPL signal (Fig. 3f) is as bright as that at 6 h decay (Fig. 3c). Neither PSL nor PSPL signals were detected on the disc decayed in direct sunlight for longer than 2.5 h (Supplementary Fig. 9), because all energy traps in the disc had already been emptied by the strong sunray stimulation.

**Other silicate-based ultraviolet-C persistent phosphors.** Besides the melilite hosts, another type of excellent silicate hosts for $Pr^{3+}$ UVC persistent luminescence is cyclosilicate, such as strontium yttrium cyclosilicate $Sr_3Y_2Si_6O_{18}$ (Supplementary Fig. 10a). The structure of $Sr_3Y_2Si_6O_{18}$ can be described as Sr/Y atom layers and $Si_3O_9$ ring layers stacking along the [101] direction; therefore, the correct formula is $Sr_3Y_2(Si_3O_9)_2$ (ref.[38]). In $Pr^{3+}$-doped $Sr_3Y_2Si_6O_{18}$, $Pr^{3+}$ ions occupy Sr/Y sites with eightfold coordination, which is quite like the surrounding of $Pr^{3+}$ in $Ca_2Al_2SiO_7$. Under excitation at 220 nm, the $Sr_3Y_2Si_6O_{18}$:$Pr^{3+}$ phosphor exhibits a strong broadband emission peaking at 266 nm (Supplementary Fig. 11a). After charged by a 254 nm lamp for 2 min, the UVC afterglow of $Sr_3Y_2Si_6O_{18}$:$Pr^{3+}$ phosphor in darkness can last for longer than 12 h (Supplementary Fig. 11b). Like $Ca_2Al_2SiO_7$:$Pr^{3+}$, the $Sr_3Y_2Si_6O_{18}$:$Pr^{3+}$ persistent phosphor also shows remarkable tagging capability in direct sunlight (Supplementary Fig. 12), room light (Supplementary Fig. 13a–e and Supplementary Fig. 14a–f) and darkness (Supplementary Fig. 14g–l), and PSL (Supplementary Fig. 13f) and PSPL (Supplementary Figs. 13g and 15) phenomena also occur in the pre-charged $Sr_3Y_2Si_6O_{18}$:$Pr^{3+}$ samples.

In addition to the melilite and cyclosilicate hosts, $Pr^{3+}$ UVC persistent luminescence was also observed in other types of silicates, including silicate garnets, oxyorthosilicates, and orthosilicates. The representative phosphors based on these three types of silicate hosts are grossular garnet $Ca_3Al_2Si_3O_{12}$:$Pr^{3+}$ (Supplementary Figs. 10b and 16), lutetium oxyorthosilicate $Lu_2SiO_5$:$Pr^{3+}$ (Supplementary Figs. 10c and 17), and lithium yttrium orthosilicate $LiYSiO_4$:$Pr^{3+}$ (Supplementary Figs. 10d and 18). Like the cases in melilite and cyclosilicate lattices, $Pr^{3+}$ ions in silicate garnet, oxyorthosilicate, and orthosilicate lattices are also highly coordinated (with eightfold coordination, sevenfold coordination, or ninefold coordination)[39–41], creating a suitably strong crystal field for $Pr^{3+}$ UVC emission.

**Ultraviolet-C persistent luminescent nanoparticles.** It is worth noting that, besides the ceramic discs and microscale powders

(via grinding), the UVC persistent phosphors can also be made in the form of nanoparticles. For instance, $Sr_3Y_2Si_6O_{18}:Pr^{3+}$ nanoparticles were synthesized using a combustion method. The nanoparticles have sizes of 100–200 nm (Supplementary Fig. 19) and the UVC afterglow can last for more than 6 h in darkness (Supplementary Fig. 20).

## Discussion

We have identified highly coordinated silicates as the suitable hosts for $Pr^{3+}$ UVC persistent luminescence and synthesized five types, more than ten kinds of $Pr^{3+}$-activated silicate-based UVC persistent phosphors. These UVC persistent phosphors can be conveniently charged by a standard 254 nm mercury lamp and their solar-blind UVC afterglow can be detected in high contrast by a corona camera in all ambient lighting conditions including in sunlight. Such a unique spectral feature enables the UVC persistent phosphors to find exciting applications in bright environments, which are not possible for visible and infrared persistent phosphors. Moreover, the long UVC afterglow, together with the PSPL capability, also enables the UVC persistent phosphors to find new applications in other important fields such as biomedicine. Finally, the requirements for host lattices for achieving $Pr^{3+}$ UVC persistent luminescence found in this study are expected to inspire the discovery of more excellent hosts for $Pr^{3+}$ ion or new UVC persistent phosphors activated by other UVC-enabling ions such as $Bi^{3+}$ and $Pb^{2+}$ (ref.[13]).

## Methods

**Fabrication of ultraviolet-C persistent phosphor discs**. The UVC persistent phosphors were synthesized using a solid-state reaction method. Take the synthesis of $Ca_2Al_2SiO_7:Pr^{3+}$ phosphor as an example, stoichiometric amounts of $CaCO_3$, $Al_2O_3$, $SiO_2$, and $Pr_6O_{11}$ source powders were ground to form a homogeneous fine powder. The mixed powder was then prefired at 900 °C in air for 2 h. The prefired material was again ground to a fine powder suitable for sintering. The prefired powder was pressed into discs with diameters of 15 and 50 mm using a hydraulic press. The discs samples were then sintered at 1300 °C in air for 2 h. The obtained ceramic discs were either used as-is, or ground to microscale (1–30 μm) fine powders for subsequent experiments. The other four representative $Pr^{3+}$-doped silicates discussed in this work were fabricated using the same procedure expect for the differences in source powders and/or sintering temperatures. The source powders and sintering temperatures are: $Sr_3Y_2Si_6O_{18}:Pr^{3+}$: $SrCO_3$, $Y_2O_3$, $SiO_2$, and $Pr_6O_{11}$ at 1250 °C; $Ca_3Al_2Si_3O_{12}:Pr^{3+}$: $CaCO_3$, $Al_2O_3$, $SiO_2$, and $Pr_6O_{11}$ at 1200 °C; $Lu_2SiO_5:Pr^{3+}$: $Lu_2O_3$, $SiO_2$, and $Pr_6O_{11}$ at 1250 °C; and $LiYSiO_4:Pr^{3+}$: $Li_2CO_3$, $Y_2O_3$, $SiO_2$, and $Pr_6O_{11}$ at 1100 °C. In all phosphors, the $Pr^{3+}$ content is 1 atom%.

**Making of ultraviolet-C persistent luminescence paints**. The UVC persistent luminescence paints were made by mixing the microscale UVC powder (30 wt.%) in acrylic polyurethane varnish. The paints were then used to write hollow or solid letters "U", "V", and "C" on a white wood board.

**Synthesis of $Sr_3Y_2Si_6O_{18}:Pr^{3+}$ nanoparticles**. The $Sr_3Y_2Si_6O_{18}:Pr^{3+}$ UVC persistent nanoparticles were synthesized using a combustion method via a highly exothermic redox reaction between nitrates and organic fuels. Two solutions containing stoichiometric amount of source precursors were prepared. One solution, called Solution A, was prepared by dissolving tetraethyl orthosilicate [$Si(OC_2H_5)_4$, 99.9%] in absolute ethanol in a crystallizing dish. The second solution, called Solution B, was prepared by dissolving strontium nitrate [$Sr(NO_3)_2$, 99%], yttrium nitrate [$Y(NO_3)_3$, 99.9%], praseodymium nitrate [$Pr(NO_3)_3$, 99.99%; the $Pr^{3+}$ content is 1 atom% in final product], and carbohydrazide (organic fuel) in a minimum amount of deionized water. Nitric acid [$HNO_3$, 70%] was added to Solution B to adjust the pH to 2–4. Solution B was then added dropwise into Solution A at 90 °C with continuous and mild stirring on a stirring hotplate to slowly evaporate the water. Once the water was evaporated, a transparent gel was formed. The gel was then transferred to a 600 °C muffle oven, where the organic fuel and metal nitrides were ignited and reacted to burn to yield white, voluminous, and foamy agglomerates of $Sr_3Y_2Si_6O_{18}:Pr^{3+}$ nanoparticles. To achieve the UVC persistent luminescence capability, the obtained product was calcined at 900 °C for 2 h in air. The calcined product was then ground to form fine nanoparticles.

**Characterization of crystal structures**. The X-ray diffraction patterns of the UVC persistent phosphors were recorded using a PANalytical X'Pert PRO powder X-ray diffractometer with Cu $K\alpha_1$ radiation ($\lambda = 1.5406$ Å).

**Characterization of spectral properties**. The spectral properties of the UVC persistent phosphors were studied using an array of spectral methods. Among them, the steady-state photoluminescence excitation and emission spectra were recorded using a McPherson spectrometer system, which comprises of a McPherson Model 234/302 excitation monochromator and a McPherson Model 2035 emission monochromator. Moreover, the steady-state photoluminescence emission spectra were also measured using a StellarNet SILVER-Nova spectrometer, where an Energetiq EQ-99X-FC broadband laser-driven light source coupled with an Edmund Optics Mini-Chrom monochromator provided the monochromatic excitation light between 220 and 260 nm. The SILVER-Nova spectrometer was also used to record the persistent luminescence emission spectra. The decay curves, PSL spectra, PSPL decay curves, and thermoluminescence curves were recorded using a Horiba FluoroLog-3 spectrofluorometer equipped with a 450 W Xe arc lamp and a R928P photomultiplier tube (185–900 nm). In these afterglow-related spectral measurements, the samples were pre-irradiated by a UVP 4-W 254 nm lamp for 2 min. For PSL and PSPL measurements, the 450 W Xe arc lamp was used to provide the 300–700 nm monochromatic stimulation light. For thermoluminescence curve measurements, the samples were heated using a homemade heating setup (temperature range, 25–280 °C; heating rate, 4 °C s$^{-1}$). In all spectral measurements, appropriate optical filters were used to avoid stray light.

**Ultraviolet-C imaging using a corona camera**. An Ofil DayCor Luminar HD corona camera was used to image and record the UVC radiation from the samples in indoor and outdoor environments. The corona camera can take bi-spectral UVC (240–280 nm)-visible images. The visible image shows the samples and the background. The UVC image shows the UVC radiation as an area of color; the color can be chosen as red, blue, yellow, etc., for the best presentation against the background. The UVC image is superimposed on the visible image; however, if the sample is far away, the UVC image tends to not exactly overlay onto the visible image because the focal planes of the UVC sensor and visible sensor are not precisely overlaid (e.g., Fig. 2f, h, Supplementary Figs. 6b, c and 12b, d). Before imaging, the samples were pre-irradiated by a UVP 4-W 254 nm lamp for 2 min. For the remote outdoor imaging experiments, a battery-powered UVP 4-W 254 nm mini lamp was used to excite the samples. For PSL imaging and PSPL imaging, a TrustFire 3800 Lumens LED flashlight were used as the stimulation source. Care was taken in imaging experiments to avoid the operators from being exposed to the UVC radiation from the glowing samples.

**Determination of ultraviolet-C persistent luminescence power**. A Newport 2936-R optical power and energy meter and a Newport 918D-UV-OD3R UV enhanced silicon photodetector were used to measure the persistent luminescence emission intensities of a decaying $Ca_2Al_2SiO_7:Pr^{3+}$ disc at different decay instants in the initial decay period (10–120 s). The measurement system has a minimum measurable power of 20 pW. The measurement setup is illustrated in Supplementary Fig. 2a and the measured intensities are shown in Supplementary Table 1. Based on the measured intensities, we used a semisphere irradiation geometry model[22] to estimate and determine the UVC persistent luminescence power intensities of the $Ca_2Al_2SiO_7:Pr^{3+}$ sample in various initial decay instants, e.g., 1, 10, 30, and 60 s, in the absolute unit of mW m$^{-2}$. The procedure used to estimate and determine the power intensities is discussed in detail in Supplementary Note 1 and Supplementary Fig. 2b, c.

## Data availability

The source data underlying Fig. 1b, c and Fig. 4a, b and Supplementary Figs. 1, 2b, c, 3, 10a–d, 11a, b, 15a, b, 16a, b, 17a, b, 18a, b, and 20a, b and Supplementary Table 1 are provided as a Source Data file. Other data that support the findings of this study are available from the corresponding author upon reasonable request.

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

## Acknowledgements

Z.W.P. acknowledges the financial support from the US National Science Foundation (DMR-1403929 and DMR-1705707) and the College of Petroleum Engineering and Geosciences, King Fahd University of Petroleum and Minerals, Saudi Arabia. F.L. thanks the financial support from the National Science Foundation of China (11774046) and the Development of Science and Technology of Jilin Province (20180414082GH). Y.F.C. acknowledges the support received from the China Scholarship Council. We thank McPherson, Inc. (Chelmsford, MA, USA) for helping measure the photoluminescence excitation and emission spectra.

## Author contributions

Z.W.P. conceived the experiments and was responsible for the project planning. X.L.W. and Y.F.C. designed and synthesized the phosphors. X.L.W., Y.F.C., and F.L. conducted spectral measurements. X.L.W., Y.F.C., and Z.W.P. carried out UVC imaging experiments. All of the authors discussed the results and contributed to the writing of the paper.

## Competing interests

The authors declare no competing interests.
