## [Peer Review File · Nature Communications]

Reviewers' comments:

Reviewer #1 (Remarks to the Author):

Comments to the Authors

Recommendation: Accept after minor revision

The manuscript entitled "Solar-blind ultraviolet-C persistent luminescent phosphor". The authors have reported the very interesting observations of persistent phosphors in ambient and dark light conditions. Indeed, authors also studied the thermal-stimulation and photo stimulation effects in bright environments, also contributes to the afterglow emission. The reported results of these phosphor materials are looks fascinating by the way the authors presented in the manuscript. However, authors need to answer the following comments, before accepting the paper.

The minor comments are outlined below:

- 1) Authors have mentioned achieving the UVC phosphor, should have high density of traps. Could the authors determine the trap density of these phosphor materials?
- 2) Would you authors confirm only the cation mismatchment is the responsible for the formation of trap states?
- 3) For the formation of trap sates, the crystal structure and composition of materials may have played a vital role. In my opinion, authors need to do provide structural details such as number of host sites for incorporating the Pr³⁺ ion.
- 4) In principle, the intensity of the long-lasting phosphors was representing in mcd/m². Could the authors measure the PL intensity of these phosphors into above-mentioned unit?
- 5) Could the authors explain what kind of defects are formed in these phosphor materials? (Either cationic or anionic defects). In my opinion formation of oxygen defects are very difficult. How could the authors get this kind of high energy defects without creating any reduction atmosphere to these samples?

Reviewer #2 (Remarks to the Author):

This manuscript reports on the persistent UVC photoluminescent properties of Pr³⁺-doped optical materials – namely Ca₂Al₂SiO₇:Pr³⁺. Uniquely, the experiments employ UV excitation using simple low-pressure mercury lamp sources to induce UVC-range persistent luminescence. This represents an interesting practical advancement, and a follow-on to the recent report by Yang et al. (ref. 23), which used X-ray excitation. However, many conclusions are not rigorously supported by the experiments, and discussion presented is overall lacking in mechanistic detail. The discussion of the potential applications of these materials is highly vague and cursory, thus the importance of this advancement is not clear. I recommend that the manuscript be revised and submitted to a materials science journal. The pdf file is also missing line numbers, which makes reviewing it a challenge. Specific comments:

-A better explanation of how such materials may be used is needed in the introduction. On page three, "non-line-of-sight optical communications" is mentioned. Explain further. "Aircraft landing aids in low visibility is mentioned"; this doesn't make sense because if visibility is low then that implies weather conditions wherein visible light is blocked, which would block UVC radiation to an even greater extent. Given the requirement to "charge" the material with a UVC light source, it seems that an actual UVC lamp could be used to much greater effect.

-The concept of downconverting (quantum cutting) phosphors mentioned on p. 3 has been made obsolete by LED light sources, including UVC LEDs.

-First sentence of second paragraph on p.4 doesn't make sense.

-The authors inaccurately state that a "small Stokes shift" is needed for effective Pr³⁺ UVC emission. Stokes shift is dependent on the chosen excitation wavelength. I believe they mean to say that the edge of the 4f5d band should not be too low in energy or else relaxation to the 3P_J states is encouraged.

-A photoluminescence excitation spectrum was not included due to instrument limitations, however, I

believe this data is important.

-The discussion of photostimulated luminescence on p. 13 could be written more clearly. At times it sounds like the visible light is exciting the phosphor, as opposed to moving pre-excited electrons out of traps to cause emission. Is there any possibility of upconversion resulting in population of traps during visible light emission? Pr³⁺ materials such as these are capable of visible-to-UVC upconversion.

-More discussion of the nature of the traps is needed. Are only oxygen vacancies playing an important role? Is it possible there are Pr⁴⁺ defects present?

-No XRD patterns are provided for the other materials. Since high phase purity is not demonstrated, the reported luminescence properties may not be representative of phase-pure materials. Overall, the sintering conditions used seem quite mild, so phase purity may be an issue.

-In the methods, the authors state "All spectra were corrected for the optical system responses". How? This is a very important aspect that should be described in detail. For example, explain how we can be certain that the wavelength dependence observed in Fig. 4a was not simply due to variability in intensity of the light source at each wavelength? Most broadband light sources equipped with monochromators have wavelength-dependent intensity. Overall, more details of the spectroscopic techniques should be included in the supporting information.

Reviewer: Ezra L. Cates

Response to Reviewers' Comments, NCOMMS-19-30591

We thank the reviewers for their constructive comments and valuable recommendations for improving the manuscript. In the revised manuscript we have fully addressed all the reviewers' comments and recommendations. The revisions are highlighted in **blue** in the revised manuscript. Below are our point-by-point responses to the reviewers' comments. With these revisions and our detailed explanation of the revisions, we hope we have removed all the reviewers' concerns.

Reviewer #1 (Remarks to the Author)

Note: The comments were divided into six parts for easy response.

Recommendation: Accept after minor revision.

Comment 1.1. The manuscript entitled "Solar-blind ultraviolet-C persistent luminescent phosphor". The authors have reported the very interesting observations of persistent phosphors in ambient and dark light conditions. Indeed, authors also studied the thermal-stimulation and photo stimulation effects in bright environments, also contributes to the afterglow emission. The reported results of these phosphor materials are looks fascinating by the way the authors presented in the manuscript. However, authors need to answer the following comments, before accepting the paper.

Response 1.1. We thank the Reviewer for the high recognition of our work and for recommending to accept after minor revision.

The minor comments are outlined below:

Comment 1.2. 1) Authors have mentioned achieving the UVC phosphor, should have high density of traps. Could the authors determine the trap density of these phosphor materials?

Response 1.2. In principle, a good persistent phosphor needs to have a significant amount of the so-called "energy traps" to store the excitation energy. The distribution of the energy traps can be measured using thermoluminescence method, as shown by the thermoluminescence curves in Figure S3; however, we are unable to determine the trap density, because we are even not clear at current stage on the nature of the traps in these new UVC persistent phosphors.

Comment 1.3. 2) Would you authors confirm only the cation mismatch is the responsible for the formation of trap states?

Response 1.3. We cannot confirm only the cation mismatch being responsible for the formation of trap states in the silicate-based UVC persistent phosphors. Besides cation mismatch, other factors may also contribute to the formation of trap states, such as various kinds of lattice defects (including oxygen vacancies) and even the inevitable small amount of silica-containing amorphous phases in these silicates. For instance, in ref. 26 the authors used electron spin-resonance (ESR) technique to examine the local structure of Ce³⁺-doped Ca₂Al₂SiO₇ violet-blue persistent phosphor (the same host as the Pr³⁺-doped Ca₂Al₂SiO₇ UVC persistent phosphor discussed in the main text) and found that oxygen vacancies were the trapping centers for electrons. Nevertheless, as stated in Response 1.2, we are not very clear on the exact nature of the trap states in the UVC persistent phosphors. Determination of the trap states in a persistent phosphor is actually one of the biggest challenge in the field of persistent luminescence; as a matter of fact, even for the most extensively studied, widely used SrAl₂O₄:Eu²⁺,Dy³⁺ persistent phosphor, people are not very clear on what the traps are. Since Reviewer #1 raised several questions around the traps, in the revised manuscript,

page 7, we acknowledged the limitations on the determination of traps in the new UVC persistent phosphors. It is our belief that some capable peers can uncover the nature of traps in these UVC persistent phosphors either experimentally or theoretically in the future.

Comment 1.4. 3) For the formation of trap states, the crystal structure and composition of materials may have played a vital role. In my opinion, authors need to do provide structural details such as number of host sites for incorporating the Pr^{3+} ion.

Response 1.4. It is no doubt that the crystal structure and composition play a vital role in the formation of trap states in a persistent phosphor. This has been proven by many visible-light and near-infrared light persistent phosphors – for a specific wavelength range, only the ones with specific composition exhibits the best persistent luminescence performance, such as $\text{SrAl}_2\text{O}_4:\text{Eu}^{2+},\text{Dy}^{3+}$ in green persistent luminescence and $\text{Zn}_3\text{Ga}_2\text{Ge}_2\text{O}_{10}:\text{Cr}^{3+}$ in near-infrared persistent luminescence. This is also true in our searching of the UVC persistent phosphors, i.e., the highly-coordinated silicate-based materials are the good hosts for Pr^{3+} UVC persistent luminescence.

The reported $\text{Ca}_2\text{Al}_2\text{SiO}_7:\text{Pr}^{3+}$ UVC persistent phosphor has a melilite structure, as shown by the XRD pattern in Figure S1. In melilite $\text{Ca}_2\text{Al}_2\text{SiO}_7$, Ca^{2+} ions are sandwiched between the layers of AlO_4 and SiO_4 tetrahedrons alternating along the c axis and are eightfold coordinated (refs. 26, 29). Each Ca^{2+} ion is bonded to four nearest neighbor O^{2-} ligand ions in the AlO_4 layer and four nearest neighbor O^{2-} ligand ions in the SiO_4 layer. The thus formed four Ca^{2+} complexes in a unit cell are structurally equivalent (ref. 26). In Pr^{3+} -doped $\text{Ca}_2\text{Al}_2\text{SiO}_7$, the Pr^{3+} ions substitute for only Ca^{2+} ions; therefore, the doped Pr^{3+} ions are eightfold coordinated. This is also the case for Ce^{3+} ions in $\text{Ca}_2\text{Al}_2\text{SiO}_7:\text{Ce}^{3+}$ (ref. 26). In the revised manuscript, we have added more information about the structure of $\text{Ca}_2\text{Al}_2\text{SiO}_7:\text{Pr}^{3+}$ (page 6).

Comment 1.5. 4) In principle, the intensity of the long-lasting phosphors was representing in mcd/m^2 . Could the authors measure the PL intensity of these phosphors into above-mentioned unit?

Response 1.5. We have measured the UVC persistent luminescence intensities of $\text{Ca}_2\text{Al}_2\text{SiO}_7:\text{Pr}^{3+}$ persistent phosphor at different decay instants in the initial 10 s to 120 s decay period using a Newport 2936-R optical power and energy meter and a 918D-UV-OD3R UV enhanced silicon photodetector. According the measured data, we estimated the persistent luminescence power intensities at 1 s and 10 s decay instants to be about $10.9 \text{ mW}/\text{m}^2$ and $4.8 \text{ mW}/\text{m}^2$, respectively. We have added the data (page 8) and the brief measurement method (page 17) in the revised manuscript. The detailed measurement method was discussed in the revised Supplementary Information, and a new Fig. S2 and a new Table S1 have been added.

It is worth noting that the unit of candela (cd; or millicandela, mcd) is only valid for the visible light that can be perceived by unaided human eyes. For the invisible UVC light, however, the unit of candela is no longer valid. Therefore, we use a unit of mW/m^2 .

Comment 1.6. 5) Could the authors explain what kind of defects are formed in these phosphor materials? (Either cationic or anionic defects). In my opinion formation of oxygen defects are very difficult. How could the authors get this kind of high energy defects without creating any reduction atmosphere to these samples?

Response 1.6. Considering the compositions and the fabrication conditions, both cationic and anionic defects can exist in the UVC persistent phosphors. In the Ce^{3+} -doped $\text{Ca}_2\text{Al}_2\text{SiO}_7$ persistent

phosphor, there exist both O^{2-} vacancies and Si^{4+} vacancies, and based on ESR results the authors found that “the energy is stored in the forms of the electrons trapped at O^{2-} vacancies such as F^+ centers and the holes self-trapped at Al^{3+} near to Si^{4+} vacancies, forming Al^{4+} ” (ref. 26). As a point defect in lattices, oxygen defects in the form of oxygen vacancies are actually inevitable in oxides synthesized by high-temperature sintering method; they can be formed, more or less, in not only reduction atmosphere but also in oxygen-rich environments. There are numerous papers discussing oxygen vacancies in oxide compounds. In persistent luminescence, oxygen vacancies are considered to be responsible for the formation of energy traps in many persistent phosphors, for instances, in $Ca_2Al_2SiO_7:Ce^{3+}$ (ref. 26), $Zn_3Ga_2Ge_2O_{10}:Cr^{3+}$ (ref. 22) and $Gd_3Ga_5O_{12}:Cr^{3+}$ (Blasse, G. *et al. J. Alloys Compd.* **200**, 17-18, 1993).

Reviewer #2 (Remarks to the Author)

Note: The comments were divided into ten parts for easy response.

Comment 2.1. This manuscript reports on the persistent UVC photoluminescent properties of Pr^{3+} -doped optical materials – namely $Ca_2Al_2SiO_7:Pr^{3+}$. Uniquely, the experiments employ UV excitation using simple low-pressure mercury lamp sources to induce UVC-range persistent luminescence. This represents an interesting practical advancement, and a follow-on to the recent report by Yang et al. (ref. 23), which used X-ray excitation. However, many conclusions are not rigorously supported by the experiments, and discussion presented is overall lacking in mechanistic detail. The discussion of the potential applications of these materials is highly vague and cursory, thus the importance of this advancement is not clear. I recommend that the manuscript be revised and submitted to a materials science journal. The pdf file is also missing line numbers, which makes reviewing it a challenge. Specific comments:

Response 2.1. We thank the Reviewer for considering our work unique and interesting.

However, we strongly disagree with the Reviewer’s claim of “many conclusions are not rigorously supported by the experiments”. We didn’t see any of the Reviewer’s comments was against any of the main conclusions of our work. As those clearly stated in the manuscript that the main conclusions of our work include: (1) the development of silicate-based UVC persistent phosphors whose UVC persistent luminescence can be induced by a standard 254 nm lamp excitation, (2) the UVC persistent luminescence emission can be sharply imaged by a corona camera in sunlight and room light, enabling these solar-blind persistent phosphors to be used as self-sustained tags in bright environments, and (3) the ambient light in the bright environments has stimulation effect on the liberation of the stored energy in the phosphors. All experiments were carefully designed and carried out to support these conclusions.

The Reviewer also asserted that “the discussion of the potential applications of these materials is highly vague and cursory, thus the importance of this advancement is not clear”. Then, what “potential applications” of our materials did the Reviewer refer to? From the questions in Comment 2.2, it is apparent that the Reviewer treated some UVC light applications (i.e., optical communication and aircraft landing aids) mentioned in the first paragraph of the Introduction section as the potential applications of our UVC persistent phosphors. How could the Reviewer treat the general introduction as the conclusion in a paper? This is simply not true – in the entire manuscript, we didn’t say a single word about the use of the UVC persistent phosphors in optical communication and aircraft landing aids (please see Response 2.2 for detail). The main potential

application of the UVC persistent phosphors discussed in the manuscript is to use them as an identification tag in bright environments, a capability that cannot be achieved by the visible-light and NIR-light persistent phosphors and therefore makes our UVC persistent phosphors unique. Therefore, it is unacceptable to us for the Reviewer using an incorrect assertion to devalue the importance of our work.

Comment 2.2. -A better explanation of how such materials may be used is needed in the introduction. On page three, “non-line-of-sight optical communications” is mentioned. Explain further. “Aircraft landing aids in low visibility is mentioned”; this doesn’t make sense because if visibility is low then that implies weather conditions wherein visible light is blocked, which would block UVC radiation to an even greater extent. Given the requirement to “charge” the material with a UVC light source, it seems that an actual UVC lamp could be used to much greater effect.

Response 2.2. One focus of this manuscript is to demonstrate the unique tagging capability of the UVC persistent phosphors in lighting conditions, particularly in daylight, a capability that is not possessed by visible-light and NIR-light persistent phosphors. Extensive UVC imaging results were shown in Figures 2&3 and Figures S4-S8 & S12-S14 and the content was comprehensively discussed in the manuscript.

In the first paragraph of the Introduction section, we gave the application background of UVC light (not our UVC persistent phosphors), including the advanced applications in optical communications and aircraft landing aids. This kind of writing is a normal practice for a research paper. Surprisingly, the Reviewer thought these were the potential applications we targeted for our UVC persistent phosphors, which are simply not true. We did not say that.

The Reviewer questioned the penetrating power of the UVC light in low-visibility weather conditions and thought its use as aircraft landing aids in low visibility “doesn’t make sense”. But that was the conclusion of several papers including refs. 11&12. Particularly, the comprehensive study in ref. 12 compared in detail the performance of visual (0.4–0.7 μm), infrared (different bands between 1 μm and 11 μm) and UVC (<0.285 μm) sensors for landing aircraft in fog. The overall conclusion of ref. 12 is “The analysis indicates that although the 1.5 micron and 3 to 5 micron IR sensors are capable of improving on the unaided eye, especially in haze and low density fog conditions, only the UV sensor, coupled with relatively minor changes to airport light lenses (to not attenuate UV light), provides the potential to aid the pilot in seeing airport lighting 3.5 times farther than the unaided eye.” For the UVC light, the authors concluded specifically as: “A sub 0.285 micron sensor like FogEye is not encumbered with background luminance, so its detection range is constant under all conditions of day and night. It shows promise to improve the pilot’s ability to acquire airport or approach lights in conditions of low visibility. It has the potential to provide the pilot with visibility enhancement equivalent to 2,400 RVR in conditions where the unaided eye would be limited to 700 RVR. This may allow for Approach Credit to permit CAT I operations in CAT IIIa conditions under a special authorization through an Operations Specification. It may also allow for Dispatch Credit to airports for which the aircraft and air crew would no longer be forecast to be below minimums because of this additional capability.”

Comment 2.3. -The concept of downconverting (quantum cutting) phosphors mentioned on p. 3 has been made obsolete by LED light sources, including UVC LEDs.

Response 2.3. The use of the term “down-conversion” is somewhat confusing. Someone treats it as “down-shifting” and someone sees it as “quantum cutting”. Our materials are down-shifting, so

to avoid further confusion, we have replaced “down-converting” using “down-shifting” in the revised manuscript (page 3).

Moreover, the concept of quantum cutting refers to the conversion of one vacuum ultraviolet photon to two or more low-energy photons in the visible and/or near-infrared spectral regions, which make luminescent materials have the quantum efficiency higher than unity. The generation of UVC light through quantum cutting has not been achieved yet. Nevertheless, the phenomenon of quantum cutting has its own merits in both fundamental research and potential applications in lighting industry, certain electronic display systems and solar cells (*Science* **283**, 663–666, 1999; *Prog. Mater. Sci.* **55**, 353–427, 2010). Therefore, it is inappropriate to compare quantum cutting with LED light sources (including UVC LEDs) and subsequently consider it to be “obsolete”.

Comment 2.4. -First sentence of second paragraph on p.4 doesn't make sense.

Response 2.4. We are wondering what is wrong with that sentence so that it was worth of being singled out by the Reviewer and was criticized as “doesn't make sense”. That sentence is a brief summary of the previous paragraph and serves as a smooth transition in context.

Comment 2.5. -The authors inaccurately state that a “small Stokes shift” is needed for effective Pr³⁺ UVC emission. Stokes shift is dependent on the chosen excitation wavelength. I believe they mean to say that the edge of the 4f⁵d band should not be too low in energy or else relaxation to the ³P_J states is encouraged.

Response 2.5. The Reviewer is incorrect on Stokes shift. According to J.R. Gispert (*Coordination Chemistry*. Wiley-VCH. p. 483. ISBN 3-527-31802-X, 2008), “Stokes shift is the difference (in energy, wavenumber or frequency units) between positions of the **band maxima** of the absorption and emission spectra (fluorescence and Raman being two examples) of the same electronic transition”. Therefore, for a specific phosphor, the Stokes shift is an intrinsic property and its value is **independent** on the chosen excitation wavelength.

It is not us who proposed the “small Stokes shift” requirement for effective Pr³⁺ UVC emission. Extensive studies were carried out in the 1980s on Pr³⁺ UVC photoluminescence. Based on their experiments, Prof. George Blasse and Dr. Alok Srivastava summarized that to ensure the occurrence of Pr³⁺ 4f¹5d¹→4f² transitions in the UVC in a solid, two general conditions are required: a small Stokes shift of less than about 3000 cm⁻¹ and an appropriate energy location of the first (lowest energy) Pr³⁺ 4f²→4f¹5d¹ excitation transition. The pioneering work can be found in ref. 15 (Prof. Blasse) and ref. 16 (Dr. Srivastava). These two general conditions were further discussed by Dr. Srivastava in refs. 18&19 and by Prof. Pieter Dorenbos in ref. 17.

Comment 2.6. -A photoluminescence excitation spectrum was not included due to instrument limitations, however, I believe this data is important.

Response 2.6. We thank the Reviewer's suggestion. We agree that a photoluminescence excitation spectrum is necessary. Now, we got the excitation spectrum of Ca₂Al₂SiO₇:Pr³⁺ phosphor with the kind help from McPherson Inc (Chelmsford, MA, USA). We have revised Figure 1b which now shows the photoluminescence excitation and emission spectra of Ca₂Al₂SiO₇:Pr³⁺ phosphor. Accordingly, we have revised the manuscript by modifying the discussion about Figure 1b in the main text (page 7), adding related description about the McPherson spectrometer in the Method section (page 17), and showing our gratitude to McPherson in the Acknowledgements section (page 24).

Comment 2.7. -The discussion of photostimulated luminescence on p. 13 could be written more clearly. At times it sounds like the visible light is exciting the phosphor, as opposed to moving pre-excited electrons out of traps to cause emission. Is there any possibility of upconversion resulting in population of traps during visible light emission? Pr^{3+} materials such as these are capable of visible-to-UVC upconversion.

Response 2.7. The photostimulated luminescence phenomenon was well documented in many literatures, including refs. 32 & 33. The photostimulated persistent luminescence phenomenon was discussed in detail in ref. 34. Nevertheless, we have added more discussion about these two phenomena in the revised manuscript (page 13).

It is highly unlikely for the visible light from xenon lamp, LED flashlight or the Sun to cause visible-to-UVC upconversion in our phosphors. Supplementary Figure S9 shows the UVC images of a bleached (i.e., all traps were emptied) $\text{Ca}_2\text{Al}_2\text{SiO}_7:\text{Pr}^{3+}$ disc being irradiated by sunlight or sunlight + LED flashlight, but no UVC signal was detected by the sensitive Ofil DayCor Luminar HD UVC corona camera.

Comment 2.8. -More discussion of the nature of the traps is needed. Are only oxygen vacancies playing an important role? Is it possible there are Pr^{4+} defects present?

Response 2.8. Several comments from Reviewer #1 are related to the trap states and we have made comprehensive responses. Please refer to Responses 1.3, 1.4 and 1.6.

Comment 2.9. -No XRD patterns are provided for the other materials. Since high phase purity is not demonstrated, the reported luminescence properties may not be representative of phase-pure materials. Overall, the sintering conditions used seem quite mild, so phase purity may be an issue.

Response 2.9. We thank the Reviewer's suggestion. The XRD patterns of other materials including $\text{Sr}_3\text{Y}_2\text{Si}_6\text{O}_{18}:\text{Pr}^{3+}$, $\text{Ca}_3\text{Al}_2\text{Si}_3\text{O}_{12}:\text{Pr}^{3+}$, $\text{Lu}_2\text{SiO}_5:\text{Pr}^{3+}$ and $\text{LiYSiO}_4:\text{Pr}^{3+}$ have been added as a new Figure S10 in the revised Supplementary Information. Accordingly, these patterns were mentioned in the revised manuscript. Moreover, a description about the X-ray diffraction measurements was added in the Method section in the revised manuscript.

Depending on materials, the sintering temperatures vary from 1100 °C to 1300 °C. These specific temperatures were determined to be the optimal ones based on our extensive material synthesis work and spectral measurements. For a specific phosphor, the same UVC persistent luminescence performance can be consistently and repeatedly obtained using the reported fabrication temperature. This range of temperatures are commonly used in the solid-state reaction fabrication of various kinds of oxide persistent phosphors, including the many near-infrared and short-wave infrared persistent phosphors developed by the authors. The temperatures of 1100 – 1300 °C are not “quite mild” by any means and we are not sure why “phase purity” may be an issue.

Comment 2.10. -In the methods, the authors state “All spectra were corrected for the optical system responses”. How? This is a very important aspect that should be described in detail. For example, explain how we can be certain that the wavelength dependence observed in Fig. 4a was not simply due to variability in intensity of the light source at each wavelength? Most broadband light sources equipped with monochromators have wavelength-dependent intensity. Overall, more details of the spectroscopic techniques should be included in the supporting information.

Response 2.10. When a spectrometer is used to measure the luminescence properties of a luminescent material, the spectral system should have correct responses to the excitation light and emission light in order to obtain the true excitation and emission spectra. Therefore, the spectrometer should be calibrated and corrected in both wavelength and intensity. Usually, the manufacturers calibrate the spectrometers in factory. But with time going, the users also need to recalibrate the system as well as the light sources. The calibration procedures can be found in instrument manuals and/or the manufacturers' websites. Above all, we don't think it is necessary to include the routine calibration details for the several spectrometers used in this manuscript. Since having a correct spectral response is a must for spectral measurement, to avoid the confusion, we have deleted that statement in the revised manuscript.

Reviewers' comments:

Reviewer #1 (Remarks to the Author):

The manuscript entitled "Solar-blind ultraviolet-C band persistent luminescent phosphor". In my opinion, Authors have answered all the questions except for the comments-1. After, careful examination of this manuscript, authors need to verify and present the data in their response sheet before its acceptance.

1) Authors may determine the trap density of this phosphor by deconvoluting the TL peak (area under the curve), gives actual trap density. Indeed, I would like to see the TL spectra of without and with doping of Pr³⁺ ions in the same experimental conditions, So that one can understand the formation of trap states either by dopant effect or complexity of crystal structure of the host composition (because the complex crystal structures can also easily form the trap states with lanthanide doping).

2) Authors also need to confirm, whether the emission originating from either Pr³⁺ doping or by formation of trap states. To examine this, authors need to measure the PL spectra of without and with doping of Pr³⁺ ions in the host material. In my opinion, the formation of anionic defects in the oxygen rich atmosphere is too difficult, unlike if the crystal structure has too complexity by its nature.

3) Authors should propose the persistence luminescence mechanism of this phosphor material. Because this material exhibits very fascinated applications which could be originated from the persistence luminescence property. So, In my opinion, it would give some better understanding to the readers.

"I am also agree with the author opinions, but still they have to clarify above mentioned comments before its acceptance".

Response to Reviewers' Comments, NCOMMS-19-30591B by Wang *et al.*

Reviewer #1 (Remarks to the Author):

Note: The comments were divided into five parts for easy response. The revisions are highlighted in blue in the revised manuscript.

Comment 1-1. The manuscript entitled “Solar-blind ultraviolet-C band persistent luminescent phosphor”. In my opinion, Authors have answered all the questions except for the comments-1. After, careful examination of this manuscript, authors need to verify and present the data in their response sheet before its acceptance.

Response 1-1. We thank the Reviewer for satisfying with our most answers and for recommending acceptance after the new questions are addressed and present. As for the previous “comments-1” (i.e., trap density), please see *Response 1-2*. The data of new experiments are present in *Response 1-2* and *Response 1-3*.

Comment 1-2. 1) Authors may determine the trap density of this phosphor by deconvoluting the TL peak (area under the curve), gives actual trap density. Indeed, I would like to see the TL spectra of without and with doping of Pr^{3+} ions in the same experimental conditions, So that one can understand the formation of trap states either by dopant effect or complexity of crystal structure of the host composition (because the complex crystal structures can also easily form the trap states with lanthanide doping).

Response 1-2. Based on our best understanding of the entire question, the Reviewer might think that the thermoluminescence (TL) spectra of $\text{Ca}_2\text{Al}_2\text{SiO}_7:\text{Pr}^{3+}$ phosphor in Figure S3 of the Supplementary Information are originated from the trap states created by not only the Pr^{3+} “dopant effect” but also the “complexity of crystal structure of the host composition”. Accordingly, the Reviewer suggested us to deconvolute the TL peak (of $\text{Ca}_2\text{Al}_2\text{SiO}_7:\text{Pr}^{3+}$, Figure S3) in order to obtain the sub-bands caused by both the Pr^{3+} dopant effect and the host effect, followed by using the area covered by the Pr^{3+} sub-band to determine the “actual trap density” of the phosphor. To verify the formation or not of trap states in the host, the Reviewer required us to fabricate un-doped host and measure its TL spectra.

Then, regarding to this question, the following three issues about persistent luminescence and the TL spectra in Figure S3 need to be clarified.

Firstly, the persistent luminescence property of a persistent phosphor is the result of delicate interplay between the emitter and the host. When studying a persistent phosphor, one cannot just talk about the host without considering the emitter. In a persistent phosphor, the emission wavelength is determined by the emitter, and the persistent emission intensity and persistence time are mainly determined by the trapping states in the host, which are generally associated with lattice defects (Yen, W. M., Shionoya, S. & Yamamoto, H. *Phosphor Handbook*, CRC Press, 2007). The traps do not emit radiation at the emitter’s characteristic wavelengths.

Secondly, the TL spectra in Figure S3 were acquired by monitoring the Pr^{3+} 268 nm emission. It was recorded as a function of the thermally stimulated Pr^{3+} 268 nm emission intensity *versus* the heating temperature (from 25° to 280 °C). It is impossible to deconvolute a Pr^{3+} TL spectrum to get a sub-band associated to the host emission, *even if* the host also emits 268 nm emission.

Thirdly, in persistent luminescence research, TL spectra are frequently used to tell the information about the distribution and the depth of energy traps in a persistent phosphor as well as the emptying process of traps in the afterglow process. The TL spectra in Figure S3 were acquired

with confined conditions: a delay time (1 min to 24 h) after the excitation was ceased, a monitoring wavelength at Pr^{3+} 268 nm emission, and in a spectrometer having a slit that limited the amount of the incoming emission light to reach the detector. Therefore, these TL spectra can only give the relative intensity of the thermally stimulated Pr^{3+} emission at 268 nm after a certain delay, and consequently, the area under one spectrum cannot be used to determine the trap density of the $\text{Ca}_2\text{Al}_2\text{SiO}_7:\text{Pr}^{3+}$ phosphor. In our opinion, in order to determine the trap density of a persistent phosphor, all energy traps in the material should be counted; that is, it needs to collect all photons at all wavelengths (~ 255 to ~ 350 nm for $\text{Ca}_2\text{Al}_2\text{SiO}_7:\text{Pr}^{3+}$ phosphor, see Figure 1 of the main text) starting from the instant the excitation was ceased. But this is very challenging in TL measurements and no such a TL measurement technique is available now. Moreover, as that we had acknowledged in the previous response letter as well as in the first revision, we are even not clear at current stage on the nature of the traps in the Pr^{3+} -doped UVC persistent phosphors, to say nothing of defining and determining the trap density.

Nevertheless, according to the Reviewer's request, we have fabricated $\text{Ca}_2\text{Al}_2\text{SiO}_7$ host without Pr^{3+} doping and measured its photoluminescence and persistent luminescence emission spectra using the same experimental conditions as that used on $\text{Ca}_2\text{Al}_2\text{SiO}_7:\text{Pr}^{3+}$. Under 254 nm light excitation at room temperature, the $\text{Ca}_2\text{Al}_2\text{SiO}_7$ host exhibited very weak photoluminescence emission in the visible spectral region (**Figure R1**); that is, unlike $\text{Ca}_2\text{Al}_2\text{SiO}_7:\text{Pr}^{3+}$ (Figure 1b in main text), no photoluminescence in the UVC spectral region was obtained in $\text{Ca}_2\text{Al}_2\text{SiO}_7$ host. After irradiation by a 254 nm lamp for 2 min, no persistent luminescence emission, including the emission in the visible spectral region, was detected in $\text{Ca}_2\text{Al}_2\text{SiO}_7$ host (**Figure R2**; the persistent luminescence emission spectrum of $\text{Ca}_2\text{Al}_2\text{SiO}_7:\text{Pr}^{3+}$ is shown in the inset of Figure 1c in main text), indicating that no apparent energy was stored in $\text{Ca}_2\text{Al}_2\text{SiO}_7$ when Pr^{3+} ion was absent. Since there is no stored energy in $\text{Ca}_2\text{Al}_2\text{SiO}_7$ host and thus no thermally stimulated emission can be monitored and recorded, *we were unable to acquire the TL spectrum of the host.*

Figure R1. Photoluminescence emission spectrum of $\text{Ca}_2\text{Al}_2\text{SiO}_7$ host under the excitation of 254 nm light at room temperature.

Figure R2. Persistent luminescence emission spectrum of $\text{Ca}_2\text{Al}_2\text{SiO}_7$ host after irradiated by a 254 nm lamp for 2 min.

Comment 1-3. 2) Authors also need to confirm, whether the emission originating from either Pr³⁺ doping or by formation of trap states. To examine this, authors need to measure the PL spectra of without and with doping of Pr³⁺ ions in the host material. In my opinion, the formation of anionic defects in the oxygen rich atmosphere is too difficult, unlike if the crystal structure has too complexity by its nature.

Response 1-3. The first sentence of this question gives us a feeling that the reviewer wanted us to confirm whether the UVC photoluminescence is originated from Pr³⁺ ion or the trap states. This feeling seems to be confirmed by the second sentence since the reviewer asked us to measure the PL (photoluminescence) spectra of the host. However, the third sentence is about the anionic defects. Then, we speculate that the reviewer might want to know whether the UVC *persistent* emission is originated from the traps created by Pr³⁺ doping or the traps created by the host lattice. Then, this question is basically similar to the first question (see *Comment 1-2*).

Nevertheless, as that described in *Response 1-2*, we have measured the photoluminescence (PL) emission spectrum of the Ca₂Al₂SiO₇ host (**Figure R1**), and no photoluminescence emission in the UVC was observed in the host.

As for the anionic defects, here the oxygen defects, created in the oxygen rich atmosphere, we had made a comprehensive explanation in *Response 1.6* of the previous response letter.

Comment 1-4. 3) Authors should propose the persistence luminescence mechanism of this phosphor material. Because this material exhibits very fascinated applications which could be originated from the persistence luminescence property. So, In my opinion, it would give some better understanding to the readers.

Response 1-4. When we were writing this manuscript, we had considered the persistent luminescence mechanism. Based on the spectral results and our understanding to persistent luminescence, we thought the persistent luminescence in the UVC persistent phosphors should share similar mechanism to many other persistent phosphors such as the Cr³⁺-activated near-infrared persistent phosphors reported by us, for which we had discussed in detail elsewhere (e.g., *Nature Materials* **11**, 58, 2012). Since there is no special new insight to be added to the mechanism, we would prefer to not repeat it in this work, but have mentioned it briefly in the revised manuscript (page 8).

Comment 1-5. “I am also agree with the author opinions, but still they have to clarify above mentioned comments before its acceptance”.

Response 1-5. We thank the Reviewer for agreeing with our opinions. Hope our new additional experiments on Ca₂Al₂SiO₇ host have clarified the Reviewer’s new comments and removed the Reviewer’s concerns.

REVIEWERS' COMMENTS:

Reviewer #1 (Remarks to the Author):

Corrections have been made to the points noted. So I think this paper should be published.

Response to Reviewers' Comments, NCOMMS-19-30591C by Wang *et al.*

Reviewer #1 (Remarks to the Author):

Comment: Corrections have been made to the points noted. So I think this paper should be published.

Response: We thank the Reviewer for satisfying with our corrections and recommending to publishing our paper.